# Validation of Polymorphisms Associated with the Immune Response After Vaccination Against Porcine Reproductive and Respiratory Syndrome Virus in Yorkshire Gilts

**DOI:** 10.3390/vetsci12040295

**Published:** 2025-03-22

**Authors:** Salvador Icedo-Nuñez, Rosa I. Luna-Ramirez, R. Mark Enns, Scott E. Speidel, Jesús Hernández, Xi Zeng, Miguel A. Sánchez-Castro, Carlos M. Aguilar-Trejo, Guillermo Luna-Nevárez, Martha C. López-González, Javier R. Reyna-Granados, Pablo Luna-Nevárez

**Affiliations:** 1Departamento de Ciencias Agronómicas y Veterinarias, Instituto Tecnológico de Sonora, Ciudad Obregón 85000, Mexico; 2School of Animal and Comparative Biomedical Sciences, University of Arizona, Tucson, AZ 85721, USA; 3Department of Animal Sciences, Colorado State University, Fort Collins, CO 80523, USA; 4Laboratorio de Inmunología, Centro de Investigación en Alimentación y Desarrollo A.C., Hermosillo 83304, Mexico; 5Zoetis Inc., VMRD Genetics R&D, Kalamazoo, MI 49007, USA

**Keywords:** antibody response, DNA, genetic markers, polymorphism, PRRS

## Abstract

Over the past 30 years, global swine producers have been battling with the porcine reproductive and respiratory syndrome (PRRS), an economically important disease caused by a virus that can infect pigs of any age. Biosecurity and vaccination are the primary strategies to control the PRRS virus (PRRSV), but the immune response against the virus and vaccines have been highly variable among animals, suggesting that a genetic component is involved in regulating such a response. Recently, genomic technology has been proposed as a tool to explore the genetic architecture related to the response to vaccination against the PRRSV. In the current study, we sequentially utilized genomic and marker-assisted technologies. This strategy allowed us to identify and further validate three polymorphisms in the genes *RNF144B*, *XKR9*, and *BMAL1* as potential molecular markers linked to the antibody response, measured as the S/P ratio, in gilts vaccinated against the PRRSV. We propose these polymorphisms as genetic markers that should be included in selection programs on pig farms aimed at enhancing the host immune response to vaccination against the PRRSV.

## 1. Introduction

The global pig industry has faced a significant challenge over the past decade due to the PRRSV. This disease results in economic losses by impairing feed conversion rates, reducing weight gain, decreasing fertility, and increasing the culling rate in sows, which currently represents a serious risk to global food security [1,2]. New knowledge about the virology, evolution, and host response to the PRRSV has increased rapidly. At the same time, new variants of the virus continue to emerge from outbreaks that have seriously hampered the effectiveness of PRRS control strategies, including vaccination [3].

Modified live virus (MLV) vaccines are commonly used to reduce the clinical impact of the PRRSV and control the dynamics of infection within herds [4]. However, several issues compromise their protective effect, including reversion to virulence, recombination between field and MLV strains, and the inability to elicit immunity against heterogeneous virus strains [5]. Another important characteristic of MLV vaccines is the variability of individual immune responses in vaccinated pigs, which suggests that the host’s genetic background may be involved in the re-emergence of outbreaks in vaccinated populations [6]. Recently, a genomic region explaining 15% of the genetic variance associated with vaccination response to the PRRSV was detected on *Sus scrofa* chromosome 7 (SSC7). This finding supports the hypothesis that the regulation of the PRRSV vaccination response is at least partially under genetic control [7].

Molecular technologies have been proposed as a potential strategy for studying the genetic architecture of complex traits in pigs [8]. Tools such as genome-wide association studies (GWASs) are used to investigate the genomic basis related to antibody response and postpartum fertility after vaccination against the PRRSV [9,10]. Likewise, GWASs have been successfully used in pigs to identify chromosomal regions harboring SNPs and candidate genes associated with the feed efficiency [11], loin muscle growth [12], intramuscular fat [13], reproductive traits [14], meat quality [15], carcass traits [16], as well as the host’s immune response to PRRSV infection and/or vaccination [17,18].

In addition to GWASs, marker-assisted selection (MAS) has been proposed as a promising strategy to improve host response after PRRSV infection or vaccination [19]. Polymorphisms in the *CD163* gene were found to be associated with resiliency to PRRSV infection, which suggests MAS can be used to control the spread of this disease [20]. Similarly, a single SNP within the gene *LONRF1* was associated with the PRRSV antibody response [21]. In another study, an SNP from the gene *EXOC4* was associated with four reproductive traits in farrowing sows, and this SNP was proposed for MAS in breeding programs [22].

Vaccination against the PRRSV is the main strategy used to control PRRS in commercial swine farms [23]. Genes associated with the major histocompatibility complex (MHC) seem to partially regulate the immune response after PRRSV vaccination. The antibody response to PRRS infection, as determined by the S/P ratio, has been linked to genetic variability in the MCH among naturally PRRSV-infected sows and F1 replacement gilts [24]. In a similar study, Serão et al. [25] reported a moderately high estimate of heritability for the S/P ratio (0.45) measured at day 46 of a PRRSV outbreak. Interestingly, these authors hypothesized that vaccination would yield similar results at the genetic level for the S/P ratio, suggesting that this variable would be a reliable indicator of the response to PRRSV vaccination.

The aforementioned molecular technologies appear to be useful in deciphering the genetic basis regulating the varied immunological response to PRRSV vaccination in gilts. Therefore, our objective was to discover and validate genetic markers associated with the antibody response, measured as the S/P ratio, after vaccination against the PRRSV in Yorkshire gilts through genome-wide and marker-assisted technologies.

## 2. Materials and Methods

The Institutional Animal Care and Use Committee of the Instituto Tecnologico de Sonora approved all procedures performed on animals in this study (approval code: 2020-0043; 1 May 2020).

### 2.1. Location and Experimental Units

The study was conducted in a full-cycle commercial swine farm located in the Yaqui Valley, Sonora, a warm and semiarid region in Northwest Mexico. The experiment included one hundred gilts (*n* = 100) from a Yorkshire maternal line, approximately 6 months old and with an initial average weight of 108.23 ± 10.9 kg, which were randomly assigned to one of two treatments, PRRSV-vaccinated (*n* = 75) and control (*n* = 25). Only healthy females with a normal physical appearance, good body condition, and previously tested as PRRSV-negative according to serological and molecular analyses were included in the study.

Gilts were housed in pens within the quarantine area, where they had free access to a water source. During the study, feed was provided in-group feed bunks and formulated according to nutritional requirements for this age and weight, as suggested to facilitate the adaptation and development of replacement breeding gilts. Briefly, gilts were fed a ration containing 3.40 Mcal/kg of digestible energy and 14.3% crude protein. The diet was composed of corn, soybean meal, whey bran, fish meal, soybean oil, minerals, vitamins, and salt fed at a rate of 2.90 kg/hd/d.

Ambient temperature (AT) and relative humidity (RH) data were collected daily from the nearest climatic station (~3 km) to the pig farm. Both AT and RH were used to calculate the temperature–humidity index (THI), according to the formula proposed by Mader et al. [26]: THI = (0.8 × AT) + [(RH/100) × (AT-14.4)] + 46.4, where AT was the ambient temperature expressed in degrees Celsius and RH was the percentage of relative humidity. The THI during the study averaged 60 units, suggesting favorable environmental conditions due to the absence of heat stress.

### 2.2. PRRSV Vaccination and Blood Sampling

After a period of adaptation (7 days), a commercial vaccine against the PRRSV was administered intramuscularly (Ingelvac PRRS MLV, Boehringer Ingelheim Animal Health Inc., Ingelheim, Germany). The day of vaccination was considered day 0 of the study. Blood samples were collected from each female on day −7 (i.e., seven days prior to vaccination) to confirm the PRRSV-negative status by real-time PCR analyses using a commercially available assay (Tetracore, Inc., Rockville, MD, USA). Blood samples were collected again on day 21 of the study and used to measure the total antibody response using a commercial ELISA kit (Idexx Labs, Portland, ME, USA). According to the manufacturer’s instructions, an S/P ratio higher than 0.4 was considered positive, as it indicated the presence of antibodies against the PRRSV. This serologic assay had 100% sensitivity and 99.5% specificity and contained both the American and European strains of the PRRS virus.

### 2.3. Genotyping and Quality Control

An additional blood sample was collected on day 40 after PRRSV vaccination and spotted on FTA cards. The cards were then sent to the Neogen-Geneseek Laboratory (Lincoln, NE, USA) and processed for DNA extraction and genotyping using a low-density SNP chip containing 10,000 SNPs (Infinium BeadChip, Illumina, San Diego, CA, USA). PLINK v1.07 software [27] was used to perform SNP quality control. For further investigation, only SNPs that met the following criteria were used in the analyses: (1) a call rate of greater than 95% or a false discovery rate below 5%, (2) a missing genotype frequency of less than 5%, (3) a minor allele frequency (MAF) above 5%, (4) a *p*-value from Fisher’s exact test for a Hardy–Weinberg equilibrium greater than 0.001, and (5) a known position or physical location within an autosomal chromosome. After quality control procedures, 8826 SNPs were retained for genomic analysis.

### 2.4. Genome-Wide Association Study

The batch effects/stratification of the test input data were corrected using the principal component analysis (PCA) option to correct for potential false positives caused by population stratification [28]. A single-locus mixed model was used to perform a GWAS to examine the relationships between the genotypes of each SNP marker and the variable S/P ratio. The additive mixed model used was y = Xb + Zg + €, where y was a vector of individual S/P ratios, X was the design matrix of fixed effects relating fixed effects in b to observations in y, Z was an incidence matrix of relating random animal additive genetic effects in g to observations in y, b was the vector of fixed effects including the gilts’ age, body weight, and the additive effect of the candidate SNP tested for association, g was the vector of random effects, and € was the vector of residual effect. It was assumed that a~N (0, Gσ^2^_a_) and e~N (0, Iσ^2^_e_), where σ^2^_a_ represents the additive genetic variance, σ^2^_e_ represents the residual variance, G was the genomic relationship matrix, and I was an identity matrix whose order is equal to the number of observations in y.

The genomic-associative analysis was performed SNP by SNP using the software SNP Variation Suite version 8.8.1 (SVSv8; Golden Helix, Inc., Bozeman, MT, USA, www.goldenhelix.com; accessed on 26 June 2024). The statistical mixed model included the age and body weight of the gilt as fixed effects, and the sire as a random term to account for family effects (relatedness). A total of 21 Yorkshire sires were included in the study.

### 2.5. Multiple Testing Correction

To account for multiple testing, we used the Bonferroni correction procedure (which assumes independence between SNPs) as a criterion to call significant associations (*p* = α/*n* = 0.05/8826 = 5.67 × 10^−6^). The experiment-wise error was 5% (α = 0.05), and the number of tests (*n* = 8826) was taken to be the number of useful SNPs. Increasing the number of SNPs tested for their association with the S/P ratio increases the risk of committing a Type I error (false positive), where an association may be discovered by chance instead of resulting from a true biological relationship. The Bonferroni correction method adjusts the significance threshold to account for the number of tests performed, ensuring that the overall Type I error rate is controlled. The conservative nature of this method, due to the assumption that the genetic variant tested is independent of the rest, reduces the potential for false positives by reducing the *p*-value for a result to be considered statistically significant, which guarantees that reported genetic associations have a low likelihood of being spurious.

Additionally, the genome-wide significance threshold for the SNP effects based on the false discovery rate (FDR) was calculated using the software SNP variation suite version 8.8.1, and a FDR < 0.01 was considered as significant. The FDR controls the expected proportion of false positives among the rejected null hypotheses, and it is a less conservative approach compared to the Bonferroni correction. Because the FDR also assumes that hypotheses are independent, an array may experience a loss of statistical power and produce false negatives if it contains a large number of SNPs in a strong linkage disequilibrium (LD) [29].

### 2.6. Functional and Enrichment Analysis

Functional pathway analysis was performed using candidate genes identified as significant on the bioinformatics online database KOBAS (http://kobas.cbi.pku.edu.cn/; accessed on 19 June 2024). The Database for Annotation, Visualization, and Integrative Discovery (DAVID, https://davidbioinformatics.nih.gov; accessed on 25 May 2024) was used to perform the gene ontology (GO) enrichment analysis and to detect the biological functions of the genes. Ensembl (https://useast. ensembl.org/index.html; accessed on 15 August 2024) and GeneCards (http://www.genecards.org; accessed on 2 September 2024) were used for gene descriptions (i.e., registration code, name, and chromosomal location). Only terms with a false discovery rate (FDR)-adjusted *p*-value < 0.05 were considered significant in this study.

### 2.7. SNP Validation Genotyping

Two independent populations (*n* = 226) composed of PRRSV-vaccinated (*n* = 140) and control (*n* = 86) Yorkshire gilts, approximately 6 months old and with an initial average weight of 105.72 ± 10.2 kg, were used to validate the SNPs discovered through GWASs as candidate markers for the PRRSV vaccination response (S/P ratio). These SNPs were rs707264998, rs708860811, rs80844350, rs705026086, and rs81358818, within the genes Ring finger protein 144 B (*RNF144B*), XK-related protein 9 (*XKR9*), carboxypeptidase Q (*CPQ*), Forkhead Box P2 (*FOXP2*), and basic helix–loop–helix ARNT like 1 (*BMAL1*), respectively.

The validation populations were located in a separate geographic area (Mayo Valley, Sonora) from the experiment GWAS population, which reduces the likelihood of shared genetic history due to migration or gene flow between them. Moreover, these populations have a distinct ancestral background because their genetic lines come from USA and Canada, which suggests that they are genetically unrelated. As part of the updating and systematization of pig programs in Sonora, the housing, nutritional management, and environmental conditions were very similar to those provided to the experiment GWAS population.

The five SNPs selected for the validation study were genotyped using the TaqMan allelic discrimination method and RT-PCR (StepOneTM, Applied Biosystems, Foster City CA, USA). The two possible SNP variants in the target template sequence were genotyped using two primer/probe pairs in each reaction. The genotyping PCR reaction was performed by adding the genomic DNA template (2 μL) plus the genotyping master mix (5 μL; Thermo Fisher Scientific, Waltham, MA, USA), genotyping custom-made assay mix (0.5 μL; probes and primers; Thermo Fisher Scientific, Waltham, MA, USA), and DNase-free water (2.5 μL).

To prepare the two negative controls, DNase-free water (2.5 μL) was added to each reaction plate instead of genomic DNA for the sample. The cycling parameters were as follows. First, denaturation was performed at 95 °C for 10 min, followed by 40 cycles of denaturation at 95 °C for 15 s and annealing and extension at 60 °C for 60 s. The StepOne Real-Time PCR System (Thermo Fisher Scientific, Waltham, MA, USA) was used to perform the PCR. Finally, the StepOne software v2.3 (Life Technologies Corporation, Carlsbad, CA, USA) was used for PCR data analysis and genotyping.

### 2.8. Statistical Analysis of the Genotype to Phenotype Validation Study

Descriptive statistics for continuous traits were calculated using PROC MEANS. Assumptions of normality of data distribution and equality of variances were tested using PROC UNIVARIATE and PROC GLM (Levene’s test), respectively. Allele and genotype frequencies were estimated using PROC ALLELE, and the Chi-square test was used to confirm the Hardy–Weinberg equilibrium. All analyses were performed with SAS software (Version 9.4; SAS Inst. Inc., Cary, NC, USA).

The SNPs identified as genomic predictors for the trait S/P ratio were analyzed using a genotype-to-phenotype associative study with a mixed-effects model. This model included the S/P ratio as the response variable, the genotype term, female age, and herd as fixed effects, body weight as a covariate, and sire as a random effect. If the genotype term was found to be an important (*p* < 0.05) source of variation in the association analyses for continuous traits, preplanned pairwise comparisons of least squares means were generated with PDIFF. These mean separation tests were executed using LSMEANS, which included Bonferroni adjustment [30]. The effects of the average allele substitution were also calculated by regressing the phenotype on the number of copies of one SNP allele as a covariate [31].

One-way ANOVA was performed to compare the S/P ratio according to the number of favorable SNP genotypes, whereas Tukey’s HSD test was used for pairwise comparisons. Significance between groups was declared at *p* < 0.05.

### 2.9. Power Analysis to Estimate Sample Size

A power analysis was conducted to estimate the sample size necessary to detect significant associations between SNPs and the trait S/P ratio. Then, a multiple regression model was performed in R, using the pwr package, with the following parameters: (1) Effect size (expected association): Small to moderate (Cohen’s f^2^ ≈ 0.02–0.15, then f^2^ ≈ 0.08); (2) Alpha (significance level): 0.05; (3) Power: 0.80 (80% chance of detecting a true effect if it exists); (4) Minor allele frequency (MAF): 0.05; and (5) Number of predictors (SNPs): 10,000. Based on these parameters, a sample size of approximately 300 individuals is recommended to detect meaningful genetic associations at 80% power with α = 0.05. Then, a total of 326 gilts were included in this study, 100 from the experiment GWAS population and 226 from the validation populations.

### 2.10. Functional Validation Using Quantitative RT-PCR

Candidate SNPs detected were validated for gene expression using quantitative polymerase chain reaction (qPCR). Oligonucleotide primers were designed from the pig genome using Primer-BLAST. The optimal annealing temperatures and primer efficiencies were determined through PCR. Primer specificity was confirmed by nucleotide sequencing of the PCR products cloned into the PCR 2.1-TOPO vector (Thermo Fisher Scientific Life Sciences, Waltham, MA, USA). cDNA was synthesized from 0.4 μg of total RNA per reaction using Superscript III (Thermo Fisher). PCR products were amplified with SYBR Green (Qiagen, Valencia, CA, USA) in an iQ5 Real-Time PCR Detection System (Bio-Rad Laboratories, Hercules, CA, USA). All qPCR results were normalized to the geometric mean of the ribosomal protein S15 reference gene. Quantitative analysis was performed using the comparative Ct method, and the fold change was calculated using the 2-∆∆CT method. Each sample was measured in triplicate to ensure the accuracy of the quantification. Resulting data were compared between control and PRRS-vaccinated pigs using Student’s *t*-test.

### 2.11. Cytokine Analysis

A commercial ELISA was used to quantify the protein levels of the porcine cytokine IFN-α in blood at 3 and 7 days after vaccination against the PRRSV. In summary, 100 μL (1.8 μg/mL) of a mouse anti-pig IFN-α antibody was utilized as the coating antibody, while a mouse anti-pig IFN-α antibody was biotinylated and employed as the secondary antibody, with recombinant porcine IFN-α (PBL Assay Science, Piscataway, NJ, USA) serving as the standard. The procedure was performed in accordance with the manufacturer’s guidelines using the provided ELISA reagents (eBioscience, San Diego, CA, USA). Student’s *t*-test was performed to compare IFN-α values between control and PRRSV-vaccinated gilts using *p* < 0.05 as the level of significance.

Pearson correlation analyses were performed to measure the association between IFN-α blood levels and gene expression (*RNF144B*, *XKR9*, and *BMAL1*) for control and PRRSV-vaccinated gilts.

## 3. Results

### 3.1. Genome-Wide Association Study (GWAS)

The PCA confirmed a homogeneous genetic background among the gilts included in the GWAS analysis. The GWAS, performed with 8826 SNPs distributed across the 18 autosomal chromosomes, discovered 10 SNPs associated with the S/P ratio, located on chromosomes 2, 3, 4, 7, 12, 13, and 18, as presented in Figure 1. All of these SNPs surpassed the correction threshold of the Bonferroni adjustment test (*p* < 5.67 × 10^−6^) and the FDR threshold (FDR < 0.01). The SNPs rs81358818, rs705026086, rs343308278, rs708860811, rs80844350, rs707264998, and rs707607708 were intronic variants located within the genes *BMAL1, FOXP2, GP9, XKR9, CPQ, RNF144B,* and *SDK1*, respectively. The SNPs rs331531082 and rs80969120 were intergenic variants located far (>0.1 Mb) from the nearest gene, whereas SNP rs3475576322 was a non-coding sequence (Table 1).

### 3.2. Biological Pathways and Gene Ontology

The pathway functional analysis detected that genes associated with S/P ratio were enriched in pathways related to (a) innate and adaptive immune system, antigen processing, ubiquitination and proteasome degradation, and class I MHC mediated antigen processing and presentation, and (b) cytokine–cytokine receptor interaction (Table 2).

These genes were significantly enriched in GO terms, including: (1) Biological processes, such as protein polyubiquitination, ubiquitin-dependent protein catabolic processes, protein ubiquitination, negative regulation of apoptotic process, engulfment of apoptotic cells, phosphatidylserine exposure on apoptotic cell surfaces, and apoptotic processes involved in development; (2) Cellular components, such as the ubiquitin ligase complex, cytoplasm, plasma membrane, and mitochondrial membrane; and (3) Molecular functions, such as ubiquitin–protein transferase activity, zinc ion binding, ubiquitin conjugating enzyme binding, and ubiquitin protein ligase activity.

### 3.3. SNP Validation Study

Of the ten SNPs associated with the S/P ratio in the GWAS, seven were located within a gene. Among these SNPs, only five met the criteria for a minor allele frequency greater than 10% (MAF > 0.10) and within the Hardy–Weinberg equilibrium (HWE, *X*^2^ > 0.05; Table 3). Therefore, these five SNPs were regarded as candidates suitable for inclusion in a genetic marker validation study.

The S/P ratio least squares means for the five SNP genotypes are presented in Table 4. The SNPs rs707264998, rs708860811, and rs81358818 were predictors of the trait S/P ratio in gilts, while the SNPs rs80844350 and rs705026086 did not show an association with this indicator of vaccination response. The genotypes exhibiting the highest favorable effects for the SNPs rs707264998, rs708860811, and rs81358818 were GG (2.51 ± 0.09), AA (2.24 ± 0.11), and CC (2.31 ± 0.08), respectively, as they demonstrated the greatest antibody response. These results suggested an important (*p* < 0.01) effect of the genes *RNF144B*, *XKR9,* and *BMAL1* on the trait S/P ratio observed in gilts included in the study.

As mentioned earlier (i.e., Table 1), the three SNPs that were found to be predictors of the trait S/P ratio are intronic variants located within non-coding regions of the genes *RNF144B*, *XKR9,* and *BMAL1*. Even though intronic SNPs do not directly code for protein synthesis, they are able to influence gene expression and function of their corresponding genes through several mechanisms such as the induction of cis-acting regulatory elements or long non-coding RNAS (lncRNAs).

### 3.4. SNP Genotype Effects

The favorable genotypes of SNP rs707264998, rs708860811, and rs81358818 showed the best antibody response in gilts. Moreover, a significant increase was observed in the S/P ratio (*p* < 0.05) as the number of favorable SNP genotypes increased. The average S/P ratio rises by approximately 0.07 and 0.20 when the number of favorable genotypes increases from 1 to 2 and 2 to 3, respectively (Figure 2).

Interestingly, higher values for the S/P ratio were observed when the favorable genotype of the SNP rs707264998 within the gene *RNF144B* was present (Figure 3), indicating this gene’s superior contribution to improve the variable S/P ratio.

Allele substitution and fixed estimated effects for the favorable SNP alleles are presented in Table 5. The SNP rs707264998 in the gene *RNF144B* had the highest allele contribution (0.301 ± 0.016) for the S/P ratio trait in gilts vaccinated against the PRRS virus. However, favorable alleles from the SNPs rs708860811 and rs81358818 also seemed to benefit this trait. Additionally, an additive fixed effect was confirmed for these SNP markers, suggesting that the sum of their individual effects was equal to their combined allele effects.

### 3.5. Quantitative RT-PCR Validation

The expression patterns of the three candidate genes associated with S/P ratio were consistent across measurements as showed in Figure 4. Their expression profiles differed between groups, confirming the normal expression in the control group versus the change in the gene expression in the PRRSV-vaccinated group. These results helped to confirm the biological significance of the SNPs from the genes identified in the current study.

### 3.6. ELISA to Measure Cytokine IFN-α

Blood values of the cytokine IFN-α at 3 and 7 days after vaccination against the PRRSV are shown in Figure 5. Non-detectable levels of IFN-α were observed in control group, whereas increasing levels of around 10 pg/mL were detected in PRRSV-vaccinated gilts. Significant differences were found between the control and PRRSV-vaccinated gilts, which confirmed the innate cytokine response against viral immunization.

### 3.7. Association Between IFN-α and Gene Expression

Pearson correlation values between serum INF-α and the expression levels of the candidate genes *RNF144B*, *XKR9*, and *BMAL1* are shown in Table 6. Individual gene expression was highly correlated with serum IFN-α values, which confirmed the associative relationship between RNAm expression and the clinical functionality of the genes significantly associated with the variable S/P ratio.

## 4. Discussion

Vaccination against the PRRSV has been one of the primary strategies for controlling the virus. However, the continuous emergence of new mutant variants and the diversity of the circulating field strains limit vaccination effectiveness [32]. Additionally, the high variability in the vaccination response among animals suggests that genetic selection in concert with an established vaccination protocol could serve as an alternative strategy to combat this disease [19,33].

Principal component analysis (PCA) was conducted to minimize confounding due to relatedness and population structure. The first few principal components typically capture the major sources of variation due to ancestry, and incorporating these components as covariates helps control for confounding from population stratification. Adjusting for central heterogeneity in GWASs allows for more accurate and reliable genetic association findings [28]. PCA results indicated a uniform genetic background among the gilts included in the GWAS analysis, suggesting a lack of population stratification. Therefore, it is less likely that population structure will obscure any association found between SNP markers and the S/P ratio [34].

Recently, genomic technology has been proposed as a tool to explore the genetic architecture associated with the response to vaccination against the PRRSV [7]. GWASs have been successfully used in pigs to identify chromosomal regions and candidate genes associated with complex phenotypes such as response to vaccination [7,18,19]. However, there are no reports on the validation of SNP markers associated with the PRRSV vaccination response in gilts through marker-assisted selection.

In the current study, we performed genomic and marker-assisted technologies sequentially to identify and further validate SNPs as potential molecular markers associated with the host immune response in gilts vaccinated against the PRRSV. After using a GWAS, we discovered 10 SNPs as predictors of a high S/P ratio. Only five of these SNPs were selected according to quality criteria and were subsequently genotyped in two independent gilt populations using TaqMan and RT-PCR molecular assays. These SNPs were tested through a genotype-to-phenotype associative statistical analysis that validated three as molecular markers for the trait S/P ratio in gilts vaccinated and non-vaccinated against the PRRSV. Our research group also successfully applied this strategy to discover and validate SNPs as markers for heat stress response in ruminants [35,36,37].

Further analyses confirmed the positive and significant effect of the favorable alleles and genotypes from the three validated SNPs. To our knowledge, this is the first report of genomic SNPs from candidate genes associated with the immune response in gilts following vaccination against the PRRSV, which were further validated as molecular markers in two independent gilt populations. According to Karlsson et al. [38] and Visscher et al. [39], due to the low likelihood that an SNP will be significant in two distinct populations, validating the SNP effects in independent animal populations appears to be the most reliable method for assessing the significance of SNPs as candidate molecular markers.

The whole genome analysis we performed identified 10 SNPs associated with the S/P ratio, which were located on chromosomes 2, 3, 4, 7, 12, 13, and 18. Of these SNPs, only five surpassed the selection criteria (i.e., MAF < 0.10 and HWE > 0.05) and were further tested for validation in an SNP-marker association study. Sanglard et al. [10] reported a genomic region on chromosome 7 (~25 Mb) in the MHC area associated with the ratio of positive samples (S/P) in sows vaccinated against the PRRSV. A novel genomic region or quantitative trait locus (QTL) for the S/P ratio was also detected on chromosomes 4, 7, and 9 using a haplotype-based GWAS, confirming the QTL previously reported in the MHC region [18]. Similarly, Hickmann et al. [40] identified a QTL within chromosomes 7 and 8 associated with the S/P ratio in Duroc and Landrace sows during a PRRS outbreak.

In our validation study, only three SNPs were identified as candidate molecular markers associated with the S/P ratio; all of them surpassed the Bonferroni adjustment threshold (*p* < 5.67 × 10^−6^) and FDR correction. These SNPs were rs707264998, rs708860811, and rs81358818, located within the genes *RNF144B*, *XKR9*, and *BMAL1*, respectively. The SNP rs707264998 had an intronic or functional position within the ring finger protein 144B (*RNF144B*) gene, which primarily expresses ubiquitin activity. Complex signaling networks are involved in the dynamic activation of the immune system following viral infection, with ubiquitination serving as a significant mechanism that modulates both the activation and repression of innate and adaptive immune responses. The *RNF144B* gene appears to play a role in the innate immune response, which acts as the first line of defense for the host against invading microbial organisms [41].

The *RNF144B* gene interacts with TANK-binding kinase (TBK1) to inactivate this enzyme, resulting in the dephosphorylation of interferon regulatory factor 3 (IRF3) and a subsequent reduction in interferon (IFN) synthesis [42]. Consequently, the *RNF144B* gene influences the precise control of IFN production, which is essential for effective pathogen clearance without harming the host [43] and for a successful vaccination response [44]. IFN is one of the cytokines that plays an important role in the response to viral infections and promotes the production of restriction enzymes that inhibit viral protein synthesis [45]. Additionally, IFN activates signaling pathways responsible for inducing the expression of interferon-stimulated genes (ISGs), which serve as potent antiviral restriction factors [44,46]. Despite the identification of more than twenty distinct IFN genes and proteins, IFN-γ is notably different due to its unique receptor activity and distinct intracellular signaling pathway [47]. IFN-γ stimulates the activation, maturation, proliferation, and expression of cytokines and effector functions of immune cells. Notably, the PI3K, MAPK/p38, and other cellular pathways exhibit extensive crosstalk with the IFN-γ signaling pathway [48].

Interferons inhibit both intracellular propagation and intercellular transmission of viruses, making them essential components of antiviral innate immunity against invading pathogens. To eliminate viral infections, type I and type II interferons (IFN-I and IFN-II) often collaborate to activate both innate and adaptive immune responses [49]. However, the expression of the *RNF144B* gene in pigs infected with the PRRSV can influence IFN production and function, facilitating the establishment of infection. *RNF144B* inhibits innate antiviral immune responses by ubiquitinating and degrading the MDA5 (melanoma differentiation-associated factor 5) and RIG-I (retinoic acid-inducible gene I) receptors, which are critical cytoplasmic RNA sensors that trigger IFN-I pathways [50]. Zhang and Feng [51] reported a variety of host miRNAs that are dysregulated and exploited by the PRRSV to evade the host innate immune response, thus promoting viral infection. During PRRSV infection, the downregulation of miR-218 increases the expression of SOCS3, which inhibits the IFN-I response and activates the NF-κB signaling pathway, leading to PRRS virus replication and persistence [52]. In PRRSV-infected pigs, *RNF144B* gene expression also affects the release of pro-inflammatory chemokines and cytokines, which are crucial for the activation of adaptive immune responses [53]. Typical immunological characteristics of PRRSV-infected hosts include sustained viremia, robust suppression of innate cytokines, delayed onset of neutralizing antibodies, induction of non-neutralizing antibodies, and dysfunction of natural killer (NK) cells [54].

The SNP rs708860811 is located in the gene *XKR9*, which encodes an apoptosis-inducing protein in cells that contain phosphatidylserine in their membranes as an “eat me” signal. This exposure can occur when cell membrane asymmetry is disrupted, such as during membrane gemmation in the release of enveloped viruses [55]. Phosphatidylserine exposure is a general characteristic of PRRSV-infected cells. A recent study demonstrated that the PRRSV exposes phosphatidylserine in its envelope, mimicking apoptosis and infecting cells via the T-cell receptor (TCR) and mucin domain-induced macropinocytosis (TIM), with CD163 serving as an alternative pathway [56]. Research involving mice has reported that all members of the XKR family, except for XKR2, are present in the plasma membrane. Transformed cells that express XKR9 respond to apoptotic stimuli with phosphatidylserine exposure, and they are efficiently engulfed by macrophages. XKR9 contains a caspase recognition site in the C-terminal region and requires direct cleavage by caspases for its function [55]. In the current study, we posited that the SNP rs708860811 in the *XKR9* gene is associated with the S/P ratio due to its potential to modulate the immune response following PRRS vaccination, as this response may be mediated by an apoptotic effect on the PRRSV induced by the vaccine.

The biological functions displayed by the *XKR9* gene include the engulfment of apoptotic cells, phosphatidylserine exposure on the apoptotic cell surface, and involvement in the apoptotic process during development. *XKR9* is implicated in apoptosis as a response to PRRS virus infection. By regulating the scrambling of phospholipids on the cell membrane, *XKR9* promotes the clearance of infected cells through programmed cell death, which serves as a form of innate antiviral defense [55]. Active scramblases provide pathways for lipid head groups to traverse the hydrophobic core of the membrane, and they are believed to function via a “credit card-like mechanism” [57]. The XKR9-regulated apoptotic process can assist dendritic cells and macrophages in presenting antigens as antigen-presenting cells (APCs). The adaptive immune system is activated when these APCs digest viral proteins and deliver them to T cells, thereby enhancing T-cell-mediated immunity against the PRRSV. Conversely, the impairment and/or death of APCs may lead to the failure of an efficient immune response due to the inability of APCs to properly activate T cells [58].

The SNP rs81358818 is located within the *BMAL1* gene, which encodes a protein involved in regulating circadian rhythms. This protein is the only component of the mammalian circadian cycle whose deletion in mouse models leads to a loss of circadian rhythms. Disruptions in circadian rhythms have been linked to the development of cardiovascular disease, cancer, metabolic syndromes, aging, and immune responses [59]. Silencing the *BMAL1* gene indicates that the proper functioning of the molecular circadian mechanism is essential for the immune response to viral infections [60]. The *BMAL1* gene product has been associated with the control of diurnal oscillations of inflammatory Ly6C monocytes in mice, a result of diurnal variations in the involved cytosines [61]. The expression patterns of the Ly6C marker in pigs suggest they are equivalent to those observed in mice. If the variations in these populations are similar in pigs and mice, it implies that receptor availability for the PRRSV in pigs may show seasonal and diurnal differences [62].

As key player in the circadian rhythm, BMAL1 influences both innate and adaptive immunity by regulating the timing of immune responses. BMAL1 is a component of the CLOCK/BMAL1 complex, which controls the production of cytokines, the activity of immune cells, and the overall cadence of immune responses [63]. BMAL1 regulates immune cell activity and the timing of cytokine production. In pigs infected with the PRRSV, BMAL1 helps modulate the circadian rhythms of macrophages and neutrophils, ensuring that the innate immune response is properly synchronized to combat the infection [64]. The CLOCK/BMAL1 complex regulates the expression of pattern recognition receptors (PRRs) involved in nucleic acid sensing during viral infections, an essential function of the innate immune system. BMAL1 is also linked to mitochondrial function and metabolic pathways that influence the activity of immune cells, thereby suppressing PRRS virus replication in infected pigs [63]. Additionally, BMAL1 controls the activation of CD8+ cytotoxic T cells, which are critical for eliminating infected cells during the adaptive immune phase. It also modulates cytokine networks that influence the humoral immune response against the PRRSV, thereby contributing to long-term immunity and protection [65].

In the current study, the SNP rs81358818 was detected as a marker for the S/P ratio in PRRSV-vaccinated gilts. This appeared to be due to a possible implication of the circadian rhythm in the immune response to immunization against the PRRSV. This suggested the involvement of seasonal and diurnal variations in the response to PRRS virus vaccination, which appeared to be genetically influenced (i.e., the *BMAL1* gene).

A more detailed analysis of the individual effects of the SNP markers revealed a greater additive contribution from the favorable SNP genotype of the *RFN144B* gene to the trait S/P ratio. Interestingly, the average value of this trait improved as the number of favorable genotypes increased, mainly due to the presence of the SNP within the *RFN144B* gene. However, according to allele substitution effects, the favorable allele of the three SNP markers validated in the current study contributed significantly to improving the S/P ratio.

Gene expression values changed in PRRSV-vaccinated gilts compared to controls, suggesting the activation of immune pathways and biological functions in response to vaccination. These include the innate immune response, antigen presenting and processing, antiviral response activation, ubiquitination and proteasome degradation, regulation of apoptotic process, and cytokine signaling. These physiological mechanisms are integral to the body’s adaptive response to the vaccine, with the primary function of providing protection against potential future PRRSV infections [66].

Serum IFN-α levels were significantly elevated following PRRSV vaccination compared to control gilts, indicating the activation of the innate immune system in response to the vaccine virus. This immune response is essential for modulating the adaptive immune system, thereby enhancing protection against future PRRSV infections. Notably, the strong correlation between IFN-α levels and gene expression suggests that the genes *RNF144B*, *XKR9*, and *BMAL1* may play a crucial role in regulating the antiviral immune response to PRRSV vaccination, mediated by interferon [67].

In the current study, three candidate SNPs were validated as potential marker predictors for the trait S/P ratio in PRRSV-vaccinated gilts. These SNPs are located within the genes *RNF144B*, *XKR9*, and *BMAL1*. Although positioned in intronic or non-coding regions, these SNPs may influence the expression of their corresponding genes [68]. This potential effect is likely due to the presence of cis-acting regulatory elements, such as transcription factors, enhancers, silencers, and insulators, which positively regulate gene expression. Additionally, long non-coding RNAs within the intronic regions may modulate protein expression through epigenetic, transcriptional, and post-transcriptional regulation [69,70].

Collectively, our results suggested that the combination of genome-wide and marker-assisted selection technologies could be an effective and beneficial strategy for identifying gilts with high S/P ratios due to their robust response to PRRS vaccination.

A limitation of this study was the use of the S/P ratio to evaluate the immune response of gilts vaccinated against the PRRSV; utilizing other parameters, such as the evaluation of neutralizing antibodies or the cellular immune response, may provide better dependent variables. However, the antibody response assessed through the S/P ratio can serve as an effective indicator trait of the ability of gilts to respond to the PRRSV vaccine. Furthermore, the S/P ratio is often a relatively simple and easily reproducible measurement that can be obtained from a standard laboratory assays, such as ELISA, a specific and sensitive test that measures antibodies against the PRRSV. Clinical indicators, neutralizing antibody titers, and cytokine profiling are all valuable measures of immune function, but they may be limited by practical challenges like labor-intensity, having enough well-trained technicians, and costs.

Serao et al. [25] reported a high heritability (h^2^ = 0.45) of the S/P ratio measured approximately 46 days after a PRRS outbreak, while Sanglard et al. [10] reported a moderate heritability (h^2^ = 0.34) of the S/P ratio in pigs vaccinated against the PRRSV. Based on these results, both groups proposed the S/P ratio to the PRRSV to be a potential and promising indicator trait for identifying genetically superior animals in their response to PRRSV vaccination.

Further studies will investigate whether the three SNP markers reported in the current study are also associated with other immunological parameters.

## 5. Conclusions

The genetic improvement of the vaccination response to PRRSV in gilts could be investigated through the integration of genomic and marker-assisted technologies. In the present study, three SNPs within the genes *RFN144B*, *XKR9*, and *BMAL1* were validated as markers for the trait S/P ratio in both vaccinated and non-vaccinated gilts. These genes are implicated in the regulation of interferons, apoptosis, and circadian rhythms, respectively, and appear to modulate the host immune response following PRRSV vaccination. We propose these SNPs as molecular markers for inclusion in swine selection programs aimed at improving the host immune response to PRRSV vaccination. However, further studies employing higher-density SNP chips, larger validation populations, and additional immunological variables are recommended to identify additional candidate genes and SNP markers within these genes.

## Figures and Tables

**Figure 1 vetsci-12-00295-f001:**
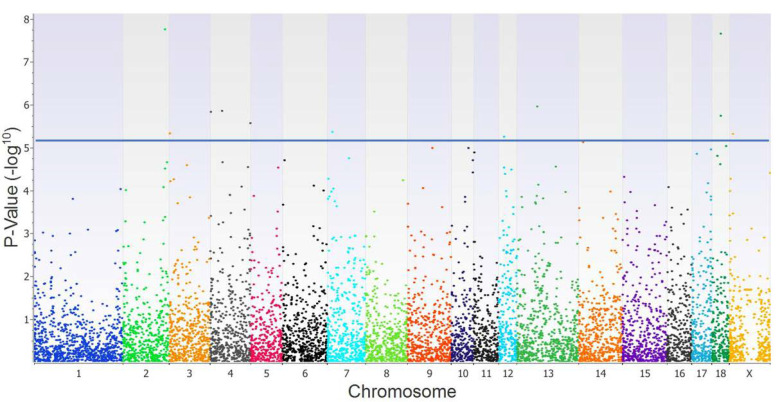
Manhattan plot showing the SNPs associated with the sample-to-positive (S/P) ratio in gilts vaccinated against the PRRSV (*n* = 100). The horizontal line corresponds to the Bonferroni adjusted threshold for a *p* < 5.67 × 10^−6^ significance level.

**Figure 2 vetsci-12-00295-f002:**
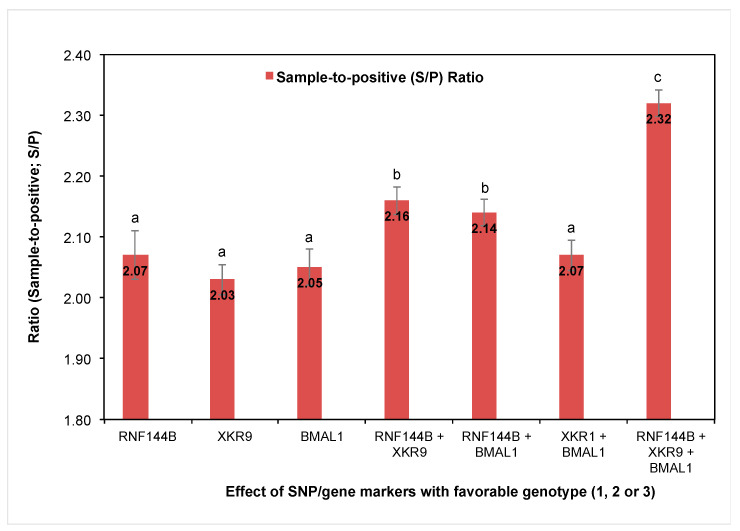
Average values (± SE) for the variable sample-to-positive (S/P) ratio according to the SNP with favorable genotypes from the genes *RNF144B*, *XKR9*, and *BMAL1* in two independent gilt populations (*n* = 226). ^a,b,c^ Indicate statistical difference among gilts with different numbers of favorable SNP/gene markers at *p* < 0.05.

**Figure 3 vetsci-12-00295-f003:**
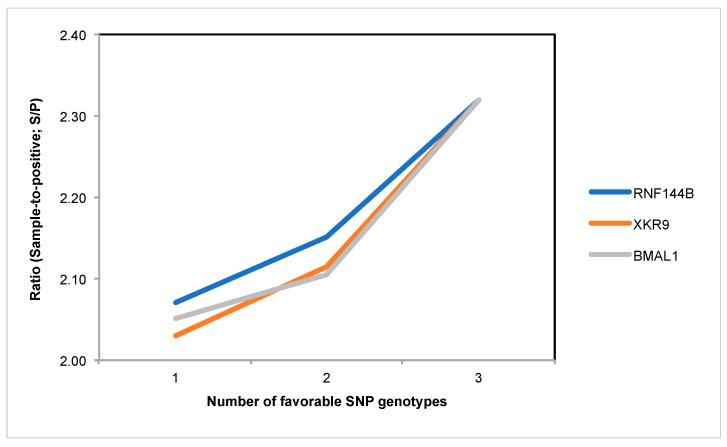
Values for the variable sample-to-positive (S/P) ratio according to the number of favorable SNP genotypes from the candidate genes *RNF144B*, *XKR9*, and *BMAL1* in two independent gilt populations (*n* = 226).

**Figure 4 vetsci-12-00295-f004:**
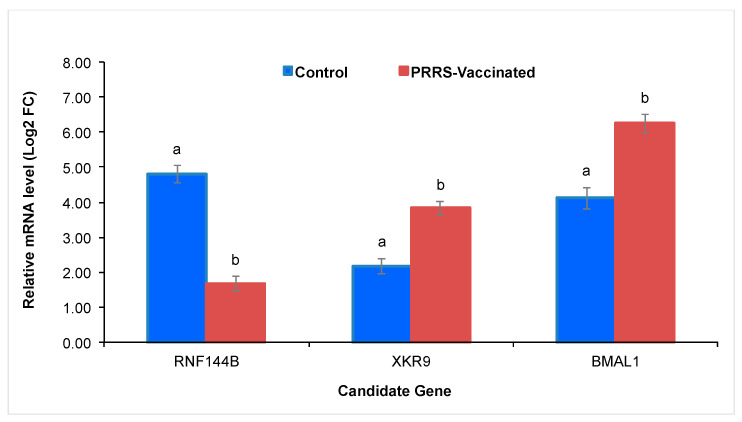
Quantitative real-time PCR (RT-PCR) validation of gene expression (i.e., *RNF144B*, *XKR9*, and *BMAL1*). Blue bars correspond to the means of mRNA expression, calculated as fold change (Log2FC) for control Yorkshire gilts (*n* = 12), and brown bars are the means for PRRSV-vaccinated Yorkshire gilts (*n* = 18). ^a,b^ Indicate statistical difference between groups at *p* < 0.05.

**Figure 5 vetsci-12-00295-f005:**
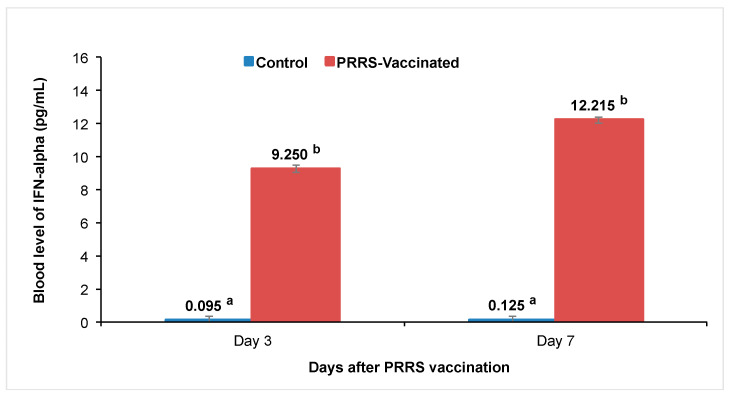
Blood levels of the cytokine IFN-α (pg/mL) between treatments at 3 and 7 days after PRRSV vaccination. Blue bars correspond to the mean values for control Yorkshire gilts (*n* = 12) and brown bars are the means for PRRSV-vaccinated Yorkshire gilts (*n* = 18). ^a,b^ Indicate statistical difference between groups at *p* < 0.05.

**Table 1 vetsci-12-00295-t001:** Summary of SNPs after *p*-value Bonferroni correction (*p* < 5.67 × 10^−6^) and FDR adjustment (*p* < 0.01) resulting from a single-marker genome-wide association study (GWAS) in Yorkshire gilts (*n* = 100) vaccinated and non-vaccinated against the PRRSV.

SNP ID ^1^	Variant ^2^	SSC ^3^	Position ^4^	Gene ^5^	Alleles ^6^	*p*-Value ^7^	FDR ^8^
rs81358818	Intronic	2	45,943,026	*BMAL1*	C/T	1.77 × 10^−8^	0.0001
rs705026086	Intronic	18	31,655,798	*FOXP2*	A/G	2.25 × 10^−8^	0.0009
rs343308278	3’UTR	13	71,562,721	*GP9*	C/T	1.12 × 10^−6^	0.0030
rs708860811	Intronic	4	64,697,573	*XKR9*	A/G	1.40 × 10^−6^	0.0025
rs80844350	Intronic	4	39,499,547	*CPQ*	A/G	1.46 × 10^−6^	0.0026
rs331531082	Intergenic	18	32,235,504	--------	C/T	1.81 × 10^−6^	0.0030
rs80969120	Intergenic	4	64,456,714	--------	C/T	2.67 × 10^−6^	0.0032
rs707264998	Intronic	7	14,111,503	*RNF144B*	A/G	4.36 × 10^−6^	0.0041
rs707607708	Intronic	3	3,156,851	*SDK1*	C/T	4.73 × 10^−6^	0.0045
rs3475576322	Non-coding	12	19,663,911	--------	C/G	5.63 × 10^−6^	0.0047

^1^ SNP reference of the Ensembl database; ^2^ SNP chromosome variant; ^3^ *Sus scrofa* autosomal chromosome number; ^4^ SNP position within the chromosome (SScrofa11.1); ^5^ candidate gene symbol (*BMAL1 =* basic helix–loop–helix ARNT Like 1, *FOXP2 =* Forkhead Box P2, *GP9 =* glycoprotein IX platelet, *XKR9 =* XK-related protein 9, *CPQ =* carboxypeptidase Q*, RNF144B =* ring finger protein 144 B, *SDK1 =* sidekick cell adhesion molecule 1); ^6^ alleles from the SNP; ^7^ SNP statistical significance; ^8^ false discovery rate.

**Table 2 vetsci-12-00295-t002:** Enriched pathways for SNP in protein-coding genes associated with S/P ratio in gilts vaccinates against the PRRSV.

Canonical Pathway ^1^	*p*-Value ^2^	Key Genes ^3^
Immune system	0.027	*RNF144B/XKR9*
Cytokine–cytokine receptor interaction	0.043	*RNF144B/BMAL1*

^1^ Significant pathways for candidate genes associated with S/P ratio; ^2^ *p*-value adjusted according to the Benjamini–Hochberg correction; ^3^ candidate gene names.

**Table 3 vetsci-12-00295-t003:** Identification, gene name, allele frequency, and Hardy–Weinberg equilibrium results for seven SNPs associated with PRRSV response as S/P ratio in Yorkshire gilts (*n* = 100).

SNP ID ^1^	Gene ^2^	Allele Frequency ^3^		HWE Test ^4^	HWE *p*-Value ^5^
		A	G		
rs707264998	*RNF144B*	0.69	0.31	0.28	0.79
rs708860811	*XKR9*	0.46	0.54	1.15	0.46
rs80844350	*CPQ*	0.51	0.49	0.86	0.58
rs705026086	*FOXP2*	0.23	0.77	0.47	0.71
		C	T		
rs81358818	*BMAL1*	0.47	0.53	0.75	0.64
rs343308278	*GP9*	0.38	0.62	24.19	< 0.01
rs707607708	*SDK1*	0.97	0.03	18.73	< 0.01

^1^ SNP reference of the NCBI; ^2^ gene symbol name (*RNF144B =* Ring finger protein 144 B, *XKR9 =* XK-related protein 9*, CPQ =* Carboxypeptidase Q, *FOXP2 =* Forkhead Box P2, *BMAL1 =* basic helix–loop–helix ARNT Like 1, *GP9 =* glycoprotein IX platelet, *SDK1=* sidekick cell adhesion molecule 1); ^3^ frequency of both alleles within gilt population; ^4^ Hardy–Weinberg equilibrium “χ^2^” test value; ^5^ “χ^2^” test *p*-value with 1 degree of freedom and α = 0.05.

**Table 4 vetsci-12-00295-t004:** Least square means ± SE according to SNP marker genotypes for the trait S/P ratio in two independent Yorkshire gilt populations (*n* = 226).

SNP ID ^1^	Gene ^2^	Least-Square Means by Genotype ± SE ^3^			*p*-Value ^4^
		AA	AG	GG	
rs707264998	*RNF144B*	1.90 ± 0.03 ^a^	2.36 ± 0.10 ^b^	2.51 ± 0.09 ^b^	0.0009
rs708860811	*XKR9*	2.24 ± 0.11 ^a^	1.88 ± 0.07 ^b^	1.76 ± 0.08 ^b^	0.0065
rs80844350	*CPQ*	1.96 ± 0.12 ^a^	1.79 ± 0.10 ^a^	1.68 ± 0.14 ^a^	0.4238
rs705026086	*FOXP2B*	1.74 ± 0.06 ^a^	2.01 ± 0.06 ^a^	2.18 ± 0.08 ^a^	0.1875
		CC	CT	TT	
rs81358818	*BMAL1*	2.31 ± 0.08 ^a^	2.14 ± 0.06 ^b^	1.87 ± 0.07 ^c^	<0.0001

^1^ SNP reference of the NCBI; ^2^ symbol name of the candidate gene (*RNF144B =* ring finger protein 144 B, *XKR9 =* XK-related protein 9, *CPQ* = carboxypeptidase Q, *FOXP2* = Forkhead Box P2, *BMAL1* = basic helix–loop–helix ARNT Like 1); ^3^ least-square means according to SNP genotype ± SE (^a,b,c^ indicate statistical difference among least-square means by genotype in the mixed model at *p* < 0.05); ^4^ *p*-value = statistical significance.

**Table 5 vetsci-12-00295-t005:** Allele substitution effect and fixed effect estimates of additive and dominance effects of the favorable allele for the S/P ratio in two independent gilt populations (*n* = 226).

SNP ID ^1^	Gene ^2^	Allele Substitution Effects			Fixed Estimates Effects		
		F. Allele ^3^	*p*-Value ^4^	Estimate ± SE ^5^	*p*-Value ^6^	AddE ^7^	DomE ^8^
rs707264998	*RNF144B*	G	<0.0010	0.301 ± 0.016	<0.0010	0.305	0.155
rs708860811	*XKR9*	A	<0.0100	0.230 ± 0.010	<0.0080	0.240	0.120
rs81358818	*BMAL1*	C	<0.0001	0.216 ± 0.012	<0.0001	0.220	0.050

^1^ SNP reference of the NCBI; ^2^ symbol name of the candidate gene (*RNF144B =* ring finger protein 144 B, *XKR9 =* XK-related protein 9, *BMAL1* = basic helix–loop–helix ARNT Like 1); ^3^ SNP allele with a favorable effect on the phenotype; ^4^ *p*-values obtained from allele substitution analysis in SAS which included the term genotype as a covariate; ^5^ estimates of the effect expressed in units of the traits ± standard error; ^6^ *p*-values for fixed effects obtained from the substitution of favorable allele analysis that included the genotype term as fixed effect; ^7^ the additive effect estimated as the difference between the two homozygous genotypes means divided by 2; ^8^ the dominance effect calculated as the deviation of the heterozygous from the mean of the two homozygous genotypes.

**Table 6 vetsci-12-00295-t006:** Pearson correlations between blood levels of cytokine INF-α and the expression values of the candidate genes associated with S/P ratio in control and PRRSV-vaccinated gilts.

Treatment	Candidate Genes		
	*RNF144B*	*XKR9*	*BMAL1*
Control	−0.6581 **	0.4738 *	0.3816 *
PRRSV-vaccinated	−0.7221 **	0.5420 *	0.4933 *

** Values are highly significant at *p* < 0.01; * Values are significant at *p* < 0.05.

## Data Availability

The data presented in this study are available on request from the corresponding author, as they belong to the records of the cooperating swine farms.

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
