# Peer review of "Validation of Polymorphisms Associated with the Immune Response After Vaccination Against Porcine Reproductive and Respiratory Syndrome Virus in Yorkshire Gilts"

_vetsci, 2025, doi:10.3390/vetsci12040295_

Round 1
Reviewer 1 Report
Comments and Suggestions for Authors
This study identified and validated three SNPs within RNF144B, XKR9, and BMAL1 genes as molecular markers for PRRS vaccine immune response in Yorkshire gilts. The research provides insights into the genetic basis of PRRS vaccination response. However, several major concerns need to be addressed.
Major comments:
1. Sample size and statistical power. With only 100 animals in the experiment population and 112 and 114 animals in the validation populations, the study appears underpowered for a genome-wide association analysis. The authors have not provided any power analysis to justify these sample sizes or demonstrate their ability to detect SNPs significantly associated with S/P ratio. To ensure the reliability of the findings, a comprehensive power analysis should be conducted, and sample sizes should be adjusted accordingly.
2. The description of population selection and validation methodology lacks crucial details. The authors have not adequately explained how the independent populations were selected or addressed potential genetic relationships between experiment and validation populations. Furthermore, there is no information regarding control for population stratification. Details on feed and environmental parameters like temperature and humidity and how feed and environmental consistency was maintained across different batches are missing. Standardization of feed, environmental factors and management practices can significantly influence the outcomes, and thus, should be clearly addressed.
3. The use of a 10K low-density chip for genotyping might limit the exploration of genetic variation. Important loci could be missed. Additionally, while PLINK software was used for SNP quality control, the manuscript does not adequately describe how confounding effects due to relatedness and population structure were addressed. Increasing the density of the genotyping chip or supplementing with whole-genome sequencing data might help in capturing a broader range of genetic variations.
4. Several critical technical errors were identified in the manuscript. Most notably, the authors used BTA (Bos taurus autosomes) to refer to Sus scrofa chromosomes in Table 1, which is a serious error. The first column of table 1 was noted as SNP reference of the NCBI, however, the NCBI SNP database currently contains only human SNP data. The description of the SNP location in the fourth column of Positions should use the thousands separator method. The SNP positions appear to be incorrect based on current genome assemblies, and the genome assembly version used for SNP annotation is not specified. For instance, the position reported for rs81358818 (49336890) conflicts with the known BMAL1 gene location in both Sscrofa11.1 (45895637-46025174bp) and Sscrofa10.2 (49303279-49335397bp).
5. The statistical methodology requires more rigorous documentation and justification. The specifications of the mixed model need more detail, and the choice of multiple testing correction methods should be better justified.
Minor comments:
1. The title of the manuscript is somewhat verbose. The term "replacement" is redundant as "gilts" inherently implies replacement females.
2. The figures in the manuscript require significant improvement in quality and clarity. For instance, Figure 1 lacks clarity, with unreadable characters in the top left corner and an apparent table symbol in the top right corner. Figure 2 shows possible duplicated bar in the RNF144B.
Author Response
We thank the Veterinary Sciences editorial office and the reviewers for all comments and suggestions that helped to improve the quality of the manuscript. We have answered to each individual comments and suggestions made by the reviewers. All revisions and new text added in the corrected version of the manuscript are highlighted in yellow.
Major comments:
- Sample size and statistical power. With only 100 animals in the experiment population and 112 and 114 animals in the validation populations, the study appears underpowered for a genome-wide association analysis. The authors have not provided any power analysis to justify these sample sizes or demonstrate their ability to detect SNPs significantly associated with S/P ratio. To ensure the reliability of the findings, a comprehensive power analysis should be conducted, and sample sizes should be adjusted accordingly.
Response: We have added in the manuscript a comprehensive power analysis with the subtitle “2.9. Power analysis to estimate sample size” (Page 6, third paragraph, lines 254 to 264).
- The description of population selection and validation methodology lacks crucial details. The authors have not adequately explained how the independent populations were selected or addressed potential genetic relationships between experiment and validation populations.
Response: We have added information about how independent populations were selected considering potential genetic relationships (Page 5, third paragraph, lines 211 to 215).
Furthermore, there is no information regarding control for population stratification.
Response: We have added information regarding control for population stratification (Page 11, second paragraph, lines 405 to 413).
Details on feed and environmental parameters like temperature and humidity and how feed and environmental consistency was maintained across different batches are missing.
Response: We have added detailed information about nutritional management (Page 3, third paragraph, lines 113 to 117) and environmental data (Page 3, fourth paragraph, lines 118 to 123).
Standardization of feed, environmental factors and management practices can significantly influence the outcomes, and thus, should be clearly addressed.
Response: We have added information regarding to standardization on nutritional management (Page 5, third paragraph, lines 215 to 218) and environmental conditions (Page 6, fifth paragraph, lines 280 to 282).
- The use of a 10K low-density chip for genotyping might limit the exploration of genetic variation. Important loci could be missed. Additionally, while PLINK software was used for SNP quality control, the manuscript does not adequately describe how confounding effects due to relatedness and population structure were addressed.
Response: We have added information regarding control for population stratification (Page 11, second paragraph, lines 405 to 413).
Increasing the density of the genotyping chip or supplementing with whole-genome sequencing data might help in capturing a broader range of genetic variations.
Response: Thank you for your suggestion. I agree that both tests are very useful as a strategy to capture a greater range of genetic variations. We did consider it, but the time available to complete the revision of the manuscript did not allow us to perform these tests. However, we were able to obtain similar results in terms of the number of significant SNPs compared to other reports where the highest density SNP-chips were used. We consider that it was helpful to have stable genetic populations, since the pig farms in the region manage very defined genetic lines.
- Several critical technical errors were identified in the manuscript. Most notably, the authors used BTA (Bos taurus autosomes) to refer to Sus scrofa chromosomes in Table 1, which is a serious error.
Response: We have made this correction in Table 1.
The first column of table 1 was noted as SNP reference of the NCBI, however, the NCBI SNP database currently contains only human SNP data.
Response: We have clarified in Table 1 that SNP references were obtained from Ensembl using the pig database (https://www.ensembl.org/index.html).
The description of the SNP location in the fourth column of Positions should use the thousands separator method.
Response: We have made this correction in Table 1.
The SNP positions appear to be incorrect based on current genome assemblies, and the genome assembly version used for SNP annotation is not specified. For instance, the position reported for rs81358818 (49336890) conflicts with the known BMAL1 gene location in both Sscrofa11.1 (45895637-46025174bp) and Sscrofa10.2 (49303279-49335397bp).
Response: We have corrected information about the SNP positions in Table 1, and clarified that the pig genome assembly version used was “SScrofa11.1” (Page 7, line 302).
- The statistical methodology requires more rigorous documentation and justification. The specifications of the mixed model need more detail,
Response: We have included more detailed specification about the statistical model (Page 4, second and third paragraphs, lines 155 to 169).
and the choice of multiple testing correction methods should be better justified.
Response: We have improved the justification for using multiple testing correction methods (Page 4, fourth and fifth paragraphs, lines 174 to 190).
Minor comments:
- The title of the manuscript is somewhat verbose. The term "replacement" is redundant as "gilts" inherently implies replacement females.
Response: We have corrected the title as suggested.
- The figures in the manuscript require significant improvement in quality and clarity. For instance, Figure 1 lacks clarity, with unreadable characters in the top left corner and an apparent table symbol in the top right corner. Figure 2 shows possible duplicated bar in the RNF144B.
Response: We improved quality in Figure 1. Also double checked Figure 2 but did not find any duplicated bar. Maybe some bar looks like duplicate but all bars have different gene, or different combination of 2 or 3 genes.
Reviewer 2 Report
Comments and Suggestions for Authors
The manuscript investigates the genetic basis of the immune response to PRRSV vaccination in Yorkshire gilts using a GWAS approach. While the findings contribute to understanding genetic regulation in vaccine responses, the study lacks critical experimental validation and detailed descriptions of the animal groups used, affecting its scientific robustness. The study mentions 100 gilts for GWAS and 226 for SNP validation but does not specify the number of animals in control vs. vaccinated groups, selection criteria, or randomization methods.
Additionally, the reliance solely on GWAS without functional validation weakens the conclusions, as no PCR/qPCR validation of gene expression, ELISA for cytokine responses, immune cell assays, and esp., clinical evaluation of the vaccination/immune responses were conducted to correlate to the biological significance of the identified SNPs. Further, the discussion does not adequately explore the mechanistic role of RNF144B, XKR9, and BMAL1 in PRRSV immunity, particularly in distinguishing between innate and adaptive immune responses. The sample size may also be a limitation, with no power calculation provided to justify its adequacy. While Bonferroni correction is applied, a False Discovery Rate (FDR) approach may be more appropriate to balance statistical stringency and SNP detection.
The manuscript would also benefit from an expanded explanation of why the S/P ratio was used as the primary immune response measure instead of more direct indicators like clinical indicators, neutralizing antibody titers or cytokine profiling.
Overall, while the study presents a novel GWAS application to PRRSV vaccine response and provides useful SNP data for marker-assisted selection, it requires major revisions. Specifically, the inclusion of functional validation experiments, clear descriptions of animal grouping, and deeper discussion on immune mechanisms would significantly strengthen the scientific rigor of the findings.
Comments on the Quality of English Language
The English could be improved to more clearly express the research and revision as requested.
Author Response
We thank the Veterinary Sciences editorial office and the reviewers for all comments and suggestions that helped to improve the quality of the manuscript. We have answered to each individual comments and suggestions made by the reviewers. All revisions and new text added in the corrected version of the manuscript are highlighted in yellow.
(x) The English could be improved to more clearly express the research.
Response: One of the coauthors, a native English speaker, have done a very detailed review of the writing in the English language and have highlighted individual grammatical corrections in yellow.
The manuscript investigates the genetic basis of the immune response to PRRSV vaccination in Yorkshire gilts using a GWAS approach. While the findings contribute to understanding genetic regulation in vaccine responses, the study lacks critical experimental validation and detailed descriptions of the animal groups used, affecting its scientific robustness. The study mentions 100 gilts for GWAS and 226 for SNP validation but does not specify the number of animals in control vs. vaccinated groups, selection criteria, or randomization methods.
Response: We have added the requested information for the GWAS experiment population (Page 3, second paragraph, lines 107 to 110) and for validation populations (Page 5, second paragraph, lines 203 to 205).
Additionally, the reliance solely on GWAS without functional validation weakens the conclusions, as no PCR/qPCR validation of gene expression, ELISA for cytokine responses, immune cell assays, and esp., clinical evaluation of the vaccination/immune responses were conducted to correlate to the biological significance of the identified SNPs.
Response: We have added method description and results from functional validation of gene expression using RT- PCR (Method: Page 6, fourth paragraph, lines 266 to 278; Results: Page 10, second paragraph, lines 384 to 395, including Figure 4).
Further, the discussion does not adequately explore the mechanistic role of RNF144B, XKR9, and BMAL1 in PRRSV immunity, particularly in distinguishing between innate and adaptive immune responses.
Response: We have added detailed information about the mechanistic role of the genes RNF144B (Page 12, third paragraph, lines 477 to 496), XKR9 (Page 13, second paragraph, lines 514 to 527) and BMAL1 (Page 13, fourth paragraph, lines 541 to 556) distinguishing between innate and adaptive immune responses.
The sample size may also be a limitation, with no power calculation provided to justify its adequacy.
Response: We have added in the manuscript a comprehensive power analysis with the subtitle “2.9. Power analysis to estimate sample size” (page 6, third paragraph, lines 254 to 264).
While Bonferroni correction is applied, a False Discovery Rate (FDR) approach may be more appropriate to balance statistical stringency and SNP detection.
Response: We have added the method of the False Discovery Rate approach (Page 4, fifht paragraph, lines 184 to 190), and results are reported in the last column of Table 1.
The manuscript would also benefit from an expanded explanation of why the S/P ratio was used as the primary immune response measure instead of more direct indicators like clinical indicators, neutralizing antibody titers or cytokine profiling.
Response: We have added the information suggested by the reviewer (Page 14, fifth and sixth paragraphs, lines 578 to 590).
Overall, while the study presents a novel GWAS application to PRRSV vaccine response and provides useful SNP data for marker-assisted selection, it requires major revisions. Specifically, the inclusion of functional validation experiments, clear descriptions of animal grouping, and deeper discussion on immune mechanisms would significantly strengthen the scientific rigor of the findings.
Response: In addition to the provided information about mechanistic roles for each gene identified in the study, we also performed a complementary functional and enrichment analysis (Page 4, sixth paragraph, lines 191 to 201) to identify pathways and biological functions in which genes associated with the S/P ratio participate (Page 7, first paragraph, lines 307 to 318).
Round 2
Reviewer 1 Report
Comments and Suggestions for Authors
I have reviewed the changes made in response to my previous comments, and I am pleased to observe that you have addressed most of the concerns effectively.
Author Response
We thank the Veterinary Sciences Editorial Office and the Reviewers for all comments and suggestions that helped to improve the quality of the manuscript. We have answered to each individual comments and suggestions made by the Reviewers. All revisions and new text added in the corrected version of the manuscript are highlighted in yellow.
Comments: I have reviewed the changes made in response to my previous comments, and I am pleased to observe that you have addressed most of the concerns effectively.
Response: Thank you very much for taking the time to review this manuscript.
Reviewer 2 Report
Comments and Suggestions for Authors
The revision address most of my concerns except those about expression validation and clinincal/functional coorelation of the significant genes, which need more improvement.
Comments on the Quality of English Language
The English could be improved to more clearly express the research.
Author Response
We thank the Veterinary Sciences Editorial Office and the Reviewers for all comments and suggestions that helped to improve the quality of the manuscript. We have answered to each individual comments and suggestions made by the Reviewers. All revisions and new text added in the corrected version of the manuscript are highlighted in yellow.
Thank you very much for taking the time to review this manuscript.
Comment: The revision address most of my concerns except those about expression validation and clinical/functional correlation of the significant genes, which need more improvement.
Response: We have added in the manuscript the information described below, in order to provide more solid foundations to improve our explanation about expression validation and clinical/functional correlation of the significant genes:
- Procedures of the ELISA clinical test performed to quantify the protein levels of the porcine cytokine IFN-α in blood (Page 6, fourth paragraph, lines 278 to 287).
- Results of the ELISA clinical test about blood serum levels of IFN-α in the experimental groups (Page 11, first paragraph, lines 418 to 423). Figure 5 (Page 11, lines 424 to 429).
- Description of the correlation analyses performed to measure the association between IFN-α blood levels and gene expression (Page 6, fifth paragraph, lines 288 to 290).
- Results of the correlation analysis performed to measure the association between IFN-α blood levels and gene expression (Page 11, second paragraph, lines 431 to 436). Table 6 (Page 12, lines 437 to 439).
- Discussion about gene expression values between PRRSV-vaccinated and non-vaccinated gilts (Page 15, fourth paragraph, lines 609 to 615).
- Discussion about measurement of IFN-α as clinical test to evaluate immune response after PRRS virus vaccination, and discussion about correlation between clinical values of IFN-α and gene expression levels (Page 15, fifth paragraph, lines 616 to 622).
Comment: The English could be improved to more clearly express the research.
Response: A second coauthor have done a very detailed review of the writing in the English language and we have highlighted individual grammatical corrections in yellow.